# A Complete Framework for a Behavioral Planner with Automated Vehicles: A Car-Sharing Fleet Relocation Approach

**DOI:** 10.3390/s22228640

**Published:** 2022-11-09

**Authors:** Asier Arizala, Asier Zubizarreta, Joshué Pérez

**Affiliations:** 1TECNALIA Research & Innovation, Basque Research and Technology Alliance (BRTA), 48160 Derio, Spain; 2Department of Automatic and Control Engineering, University of the Basque Country (EHU/UPV), 48013 Bilbao, Spain

**Keywords:** automated vehicle, platooning, behavioral planner, 3D environment

## Abstract

Currently, research on automated vehicles is strongly related to technological advances to achieve a safe, more comfortable driving process in different circumstances. The main achievements are focused mainly on highway and interurban scenarios. The urban environment remains a complex scenario due to the number of decisions to be made in a restrictive context. In this context, one of the main challenges is the automation of the relocation process of car-sharing in urban areas, where the management of the platooning and automatic parking and de-parking maneuvers needs a solution from the decision point of view. In this work, a novel behavioral planner framework based on a Finite State Machine (FSM) is proposed for car-sharing applications in urban environments. The approach considers four basic maneuvers: platoon following, parking, de-parking, and platoon joining. In addition, a basic V2V communication protocol is proposed to manage the platoon. Maneuver execution is achieved by implementing both classical (i.e., PID) and Model-based Predictive Control (i.e., MPC) for the longitudinal and lateral control problems. The proposed behavioral planner was implemented in an urban scenario with several vehicles using the Carla Simulator, demonstrating that the proposed planner can be helpful to solve the car-sharing fleet relocation problem in cities.

## 1. Introduction

Population growth in urban locations has created several transportation-related issues. Road congestion and parking spot shortage, as well as the maintenance costs [1] are changing the traditional vision of becoming a car owner. However, there is still an appreciation for the flexibility a private car gives in comparison to public transport. The results of the Central Statistics Office show that almost 75% of the people in Ireland use a bus less than once a month [2], being the main reason for the lack of service or inconvenience of the stops. In this context, car-sharing services have emerged as a new business model, which combines the benefits of private transport and the elimination of parking and maintenance management for the driver [3,4].

The car-sharing model requires managing a fleet of vehicles in urban environments. Typically, vehicles are parked in several reserved spots around a city (“one-way-trip”) [5], so that vehicles can be rented and parked in another spot after the renting period has ended. However, this leads to imbalances between parking spots, requiring approaches to reallocate the fleet. Reallocation is carried out by staff, which manually move a vehicle from one spot to the other [6,7]. Hence, the optimization of this process is critical for reducing costs [8]. In this field, most of the works are focused on the optimization of the trajectories to be performed by the staff [9,10], while other works have proposed to optimize the number of workers, by using trucks [11]. However, this is still an open research area.

In this context, automated vehicles (AVs) offer a potential framework for the development of new shared vehicle fleet management approaches. Among the different strategies proposed for AVs, platooning approaches [12] fit the aforementioned car-sharing fleet management applications. In particular, an approach in which a leader vehicle, driven by a human, can be followed by a set of AVs would allow reducing the costs related to fleet reallocation. Moreover, the automation level of commercial vehicles has increased in the last few years, and Vehicle-to-Vehicle (V2V) communications have been proposed [13], providing a technological basis for this approach. However, to the best of the author’s knowledge, there are no works proposed in the literature related to the implementation of platoon-based fleet-relocation approaches in urban scenarios for car-sharing applications. This fact will be further analyzed in the rest of this section.

For an urban-scenario-based car-sharing application, a behavioral planner is critical, as it determines which maneuver has to be executed at each instant: car following, merging with the platoon, parking, or de-parking. In the literature, two main approaches have been proposed for implementing behavioral planners in AVs: Finite State Machines (FSMs) and Neural Networks (NNs). An FSM is a rule-based approach that alternates between different trajectory planners, each of them being specific for a particular situation or state [14,15]. The difficulty of designing an FSM is related to the definition and transition between states, especially when dealing with complex scenarios. On the other hand, neural-network-based approaches use previously defined training data to generate trajectories from the sensor’s raw data [16]. However, these approaches present issues when exploring unknown or complex scenarios. To solve them, Reinforcement Learning (RL) approaches have been used to aid in the training process [17,18]. In all approaches, the amount of states available in each application is directly connected to the control goal; in most of these applications, the states are limited to a set associated with SAE Level 5: vehicle following, parking/de-parking, or merging, which will be analyzed next.

Vehicle following in platooning is an area in which many contributions have been proposed in the last few years, although most of them have been focused on inter-urban or highway scenarios, in which external elements are scarce and the tracking control problem can be simplified [19,20,21,22,23,24,25,26]. In particular, one of the main goals of platooning has been focused on increasing fuel efficiency [20], which is achieved by reducing the air drag as much as possible by maintaining a close, but safe inter-vehicle distance by generating optimal vehicle control inputs. Works focused on urban scenarios are scarce [27,28,29], due to the increased set of situations that can arise (e.g., intersections, traffic lights, lane changes due to parked vehicles, pedestrian interruptions, etc.).

Automatic Parking (AP) is a function already implemented in some vehicles. However, there is still room for improvement in this area. The AP problem can be divided into three phases: parking spot detection, path planning, and control. Both low-speed control and detection problems are well studied and not specific to the AP function, so most of the articles related to AP are focused on solving the path planning problem. In fact, those articles often make use of simple control algorithms, such as fuzzy logic [30,31], bang-bang [32,33], or PID strategies [34]. Nevertheless, recent works focused on the control of the trajectory tracking usually make use of Nonlinear Model Predictive Control (NMPC) [35,36], which gives the possibility to use output and state constraints.

Early approaches propose trajectories based on geometrical curves [33,37]. Although their application is easy to implement, they have some limitations. They require the vehicle to be in a specific initial region [37] and do not work well in narrow spaces. To solve this problem, well-known path planning algorithms are used in addition to kinematic constraints due to the nonholonomic nature of the vehicle. In [38], a directional Hybrid A* algorithm was proposed to lead the vehicle to the initial position, and then, a geometric curve was used for the parking path planning. One of the main concerns regarding this method, however, is the high computational cost and, therefore, the incapability of recreating a new path dynamically. Rapid Random Trees (RRTs) based path planners have been proven to be able to generate a path from any starting point, allowing both forward and backward movements [39,40,41,42,43,44] and with a reduced computational cost at the price of reducing trajectory efficiency.

Finally, merging or joining strategies in platooning scenarios are usually cooperative maneuvers that let two vehicles agree on which position the entering vehicle should enter and generate a trajectory according to the decision made. Most works in this area focus on merging scenarios at highway incorporations [27,45], focusing on lateral path planning [46] or even negotiation with communication V2V [47].

The aforementioned maneuvers have been studied and proposed for specific applications. In the case of platooning-related works, most focus on inter-urban or highway scenarios. However, few works in the literature focus on urban scenarios, and to the best of the author’s knowledge, there is a lack of literature related to the application of these approaches to car-sharing applications in urban environments. Therefore, this work aims to provide insight into this area, by means of the following contributions:A behavioral planner framework for the car-sharing relocation problem using platooning in urban scenarios.The definition of a basic V2V protocol integrated into the framework specifically designed for car-sharing applications.An adapted urban testing environment using the Carla Simulator for the specified use case.

The proposed framework considers a scenario in which a human drives a leading vehicle and gathers a set of AVs, which make use of the proposed behavioral planner to relocate them to urban environments.

This paper is structured as follows. In Section 2, the proposed frameworks, the behavioral planner, and the implementation of each maneuver is given. Section 3 describes the simulation tools used for the validation process. Section 4 shows the results of the simulations obtained when executing the proposal. Finally, in Section 5, some conclusions are gathered, as well as future work planned.

## 2. Behavioral Planner Framework

### 2.1. Overview

The approach proposed in this work is focused on car-sharing applications, in which a human driver leads a leader vehicle and relocates a set of AVs, which will be used to provide car-sharing services. To define the behavioral planner, first, a brief explanation of the cycle of the relocated AVs is detailed.

Figure 1 summarizes the process of relocating vehicles. First, a set of vehicles is assumed to be parked in predefined spots, so that a global planner can use these spots, as well as the information of the map, or even the current traffic, to define a global path that allows the relocation of the different vehicles and that will be provided to the driver of the leader vehicle. Please note that the definition of this global planner is not the focus of this work and that the optimal path will be considered known in the use case.

When the leader vehicle (in blue) approaches the parking spot of the AV (1), a signal is sent to the AV (in orange) to start the platoon merging process. For that purpose, the AV will perform a de-parking maneuver, followed by a merging maneuver. Once the AV has merged with the platoon, it operates in vehicle-following mode (2). The platoon will continue to pick up the different vehicles until the relocation parking spot defined by the global planner is approached (3). In this case, the AV (in orange) will perform a de-merging maneuver, followed by a parking maneuver.

As the aforementioned functionality is well structured and the transition flags are clear, a Finite State Machine (FSM) is defined, in which each maneuver is defined using a state, as can be seen in Figure 2. There are five maneuvers a relocated vehicle can be in: platoon-following, parking, waiting, de-parking, and joining.

The aforementioned state machine is designed to reproduce the cycle of AVs in car-sharing applications. To implement it, basic V2V communication was assumed to exist between the leader and the follower AVs. This way, the initial state of an AV is waiting in a parking spot (Figure 1 (1)), broadcasting a communication message containing its position, state, and ID. Once the leader approaches, it will receive the broadcast message and will communicate with the AV if joining is possible, determining a platoon position at the end of the platoon. Note that joining may not be possible if other vehicles are blocking the AV for the de-parking maneuvers.

In order to join the platoon, the AV enters the de-parking state and will orient itself with the nearest lane. Depending on the relative position for the rest of the platoon, a joining process will be required (if the platoon is in another lane or the distance to the platoon is more than the desired platoon inter-vehicle distance), activating this state.

Once the AV achieves to the desired distance to the last member of the platoon, the vehicle is considered merged, and the follower AV sends a message to the leader so that the platoon list can be updated. Then, the AV enters the following state (Figure 1 (2)). In this state, each AV calculates its control values, making use of the position and velocity of the preceding vehicle. When the desired parking spot is reached (Figure 1 (3)), the leader will signal the last vehicle on the platoon to park, by entering the parking state and jumping to the waiting state once finished to start the cycle again.

To implement the aforementioned approach, it was assumed there exists connectivity between the leader and the AV followers at small inter-vehicle distances. The literature has proven that Vehicle-to-vehicle (V2V) communication can increase string stability if shared information is used for the longitudinal and lateral control [48].

In this regard, the proposed approach considers that all vehicles share information (Figure 3). The information flow topology is predecessor–leader following [49], which means every vehicle in the platoon will receive information from both the leader and their predecessor. Additionally, the last vehicle shares its position and velocity values with the leader so it can manage the traffic light situations.

The proposed basic messaging system is centralized by the leader, which is responsible for managing the platoon. Since, in this case scenario, the leader of the platoon is driven by a human driver, possible wrong decisions due to communication mistakes are not considered in this paper. Messages are sent broadcast with enough information for each follower to interpret. Each message contains the following information:Vehicle state: This number represents if the follower is in the platoon or parked in a parking spot.Platoon position: This is the relative position of the vehicle in the platoon.Parking spot: If an AV follower is to be parked, the leader of the platoon sends the position of the parking spot to the assigned AV follower.

Next, based on the defined behavioral planner and considering the communications framework, the different maneuvers will be detailed. Note that, in the proposed study case, each maneuver can implement a particular control strategy that best fits the requirements of the maneuver. Although the development of particular control approaches is not in the scope of this paper, a comparative study was carried out using different control configurations to define the one selected for the study case implemented in this work. This way, three of the most-used approaches in the literature were evaluated:PID: Longitudinal and lateral controls were considered independent, using a fixed velocity reference for the longitudinal control and using a PID-based control for both controls.Longitudinal PID and lateral MPC: Longitudinal and lateral controls were considered independent. A fixed speed reference was used for the longitudinal control, in which a PID was implemented, while a bicycle-model-based MPC was used for steering.MPC: Both the speed reference and steering input were generated by a single MPC, which considers a bicycle model for its prediction. The MPC generates the velocity reference, which is followed by a PID.

The results are shown in Table 1, where the Root Mean Square (RMS) of the lateral error was selected as a performance metric. Figure 4 shows the trajectory performance of each controller for the two scenarios evaluated (parallel and battery parking). The controller parameters evaluated are detailed in the caption.

Based on the aforementioned table and considering the particular study case analyzed in this work, the control approaches detailed in Table 2 were selected to implement the maneuvers. Please note, however, that this does not imply that the proposed approach is limited to these controllers, as it can consider different ones. More detail regarding each controller will be given in the next sections.

### 2.2. Platoon Following

In the platoon-following state, the AV followers have to track the trajectory defined by the preceding vehicle. In the literature, this approach is implemented by considering two controllers [50]: longitudinal and lateral control. The first one generates the inputs for the throttle and brake, and it is responsible for the longitudinal acceleration and extension of the speed of a vehicle. The second controller generates the steering inputs to follow a specific path.

In the following subsections, the controllers chosen for the platoon-following maneuver will be explained in further detail.

#### 2.2.1. Longitudinal Control

This area has been thoroughly studied in the literature, with two major research lines: ensuring string stability, that is ensuring that the distance between the platoon members remains bounded and stable; and optimizing fuel consumption. The first is typically implemented using controllers that do not require the model of vehicles, such as PID-based approaches [25,51], while the latter requires considering the model of the vehicles to perform the minimization. In these cases, Model-based Predictive Controllers (MPCs) are commonly used [52,53,54].

Traditionally, vehicle following control is labeled as Adaptive Cruise Control (ACC) in the literature. This is meant to be an evolution of the classic Cruise Control (CC) technology, which uses an estimation of the velocity of the preceding vehicle to generate a speed reference for the controller to follow. If V2V communication is available between vehicles, more accurate data can be used, provided by the preceding vehicle, and longitudinal control is labeled as Cooperative Adaptive Cruise Control (CACC), which is the strategy implemented in this work.

Since urban scenarios are dynamic and many decisions have to be made during a single driving session, a string-stability-oriented solution was chosen, where a PID-based CACC is used to generate a velocity reference for the lower PID-based level controller to act upon (Figure 5). Therefore, the control law follows the classic formulation:(1)$$v_{ref}\left(t\right)=K_{p}e\left(t\right)+K_{i}\int_{0}^{t}e\left(t\right)\;d\tau +K_{d}\frac{de(t)}{dt}$$
where $v_{ref}$ is the reference velocity for the lower controller, $K_{p}$, $K_{i}$, and $K_{d}$ are adjustable parameters, and *e* is the predicted error of the distance between the controlled vehicle and the previous one:(2)$$e=d_{pred}-d_{wanted}$$

The predicted distance makes use of the information from the previous vehicle and assuming the acceleration will maintain constant through the next time step:(3)$$d_{pred}=d+\left(v_{p}-v\right)dt+\frac{a_{p}-a}{2}dt^{2}$$
where *d* is the distance between the controlled and preceding vehicles along the trajectory, *v* and $v_{p}$ are the longitudinal velocities of the controlled and the previous vehicles, respectively, and *a* and $a_{p}$ are the longitudinal accelerations for the controlled and the previous vehicles, respectively.

#### 2.2.2. Lateral Control

When it comes to lateral control in AVs, the main goal of the problem is to follow a trajectory defined previously. Perception systems provide the information needed for this task. This procedure can be applied to several AV functionalities, such as lane-keeping, lane changing, or obstacle avoidance. In this work, the reference trajectory is defined by the leader vehicle, so the followers are controlled to maintain the same track as the leader. The most common approaches in the literature are PID- [55,56], fuzzy- [30,31], and MPC-based [57,58] controllers. However, MPC controllers seem to be more predominant thanks to the possibility of introducing vehicle dynamics and safety and comfort constraints into the control law.

In urban scenarios, a traditional assumption in the literature is to neglect tire slip due to low lateral accelerations. Hence, a model-based predictive control based on the so-called bicycle kinematic model of the vehicle can be used as the lateral controller (Figure 5):(4)$$\begin{array}{ccc} x_{k+1} & = & x_{k}+v_{k}cos\left(\phi_{k}\right)T_{s}\\ y_{k+1} & = & y_{k}+v_{k}sen\left(\phi_{k}\right)T_{s}\\ \phi_{k+1} & = & \phi_{k}+\frac{v_{k}}{L}tan\left(\delta_{k}\right)T_{s} \end{array}$$
where $x_{k}$ and $y_{k}$ are the coordinates of the center of the rear axle of the vehicle at the k∈[0,h]th time step represented in the reference system of the vehicle at time k=0, *h* being the prediction horizon; $v_{k}$ is the longitudinal velocity, which is considered constant within the horizon; $\phi_{k}$ is the yaw angle; *L* is the wheelbase of the vehicle and $T_{s}$ the step time. $\delta_{k}$ is the equivalent steering angle in radians transported to the center of the front track for a supposed 100% Ackermann configuration.

If $\Delta u^{+}={\left[\begin{array}{ccc} \Delta \delta_{k} & \ldots & \Delta \delta_{k+h-1} \end{array}\right]}^{T}$ is the controller output and $\hat{x}\left(k\right)$ is the state of the system, as defined in Equation (Equation 4):(5)$${\hat{x}}_{k}={\left[x_{k},y_{k},\phi_{k}\right]}^{T}$$
then the cost function to be optimized by the MPC is defined as follows:(6)$$J={\left(\hat{y}-y_{\mathrm{ref}}\right)}^{T}Q\left(\hat{y}-y_{\mathrm{ref}}\right)+\Delta u^{+T}R\Delta u^{+}$$
where $\hat{y}$ is the *y* coordinate of the vehicle along the horizon *h* represented in the vehicle’s initial state reference system and $y_{\mathrm{ref}}$ is the y coordinate of the reference trajectory represented in the same reference system. Q and R are the weighting matrices used to tune the controller once again. Steering input is limited to st<0.7 rad and st>−0.7 rad due to physical restrictions.

The lateral reference is generated from the positions shared periodically by the leader. For that purpose, the last points provided by the leader are used to calculate the local path to be followed. This is better explained in Figure 6 with a simplified case using only three points. As can be seen, at each iteration, a new point is added to the path if the ego vehicle has traveled a certain distant from the last point. At the same time, the oldest point in the list is removed if it is behind the controlled vehicle. The reference for the controller is achieved by obtaining equidistant points at $d=v\;T_{s}$, where *v* is the current longitudinal speed of the vehicle and $T_{s}$ the control period.

### 2.3. Parking/De-Parking

The parking or de-parking maneuver will be executed in the states with the same name and will allow picking (or leave) a follower in the platoon. A geometric approach to generate a path to the parking spot is used, which is combined with an MPC approach to track the defined trajectory, as explained next.

Note that only parking will be detailed, as de-parking will consist of an identical approach, but in reverse order.

#### 2.3.1. Path Generation

To define a path to park a follower, two scenarios were considered: battery parking, which considers all cases in which the orientation of the parked vehicle is not parallel with the orientation of the road; parallel parking, in which the parking spot is parallel with the road.

When considering battery parking, the typical human driver maneuver considers: (1) lining up the vehicle with the parking spot from the initial position ($P_{ini}$); (2) guiding it in a straight line to reach the final position. Geometrically, the optimal way of generating the described trajectory is by combining an arc tangent to the initial and the final orientation axes, followed by a straight line between the tangent point ($P_{tang})$ and the final position ($P_{end}$).

Based on this behavior, a geometric approach has been implemented for battery parking, which is summarized in Figure 7, in which two different trajectories are defined (green and orange). In addition, to define the trajectories, several constraints have to be considered: the steering angle needed to perform the curve must be within the boundaries of the controlled vehicle; no collision should happen with the nearby obstacles (vehicles, pedestrian, trees, etc.).

This last consideration is ensured by checking the intersection between a bounding box around any obstacle and the controlled vehicle. To do so, additional paths are generated from each vertex of the bounding box around the controlled vehicle. Then, the intersection between them and the edges of the bounding boxes surrounding the obstacles is checked (Figure 8. If an intersection is confirmed, the trajectory generated is discarded. This method allows the application of a Safety Coefficient (SC) to offline collision checking to enhance the robustness of the approach. Therefore, an SC of δ=1.05 was used in the testing simulations. Notice that any number of bounding boxes can be added to the trajectory generator, even as a method to avoid invading some spaces, such as the opposite direction lane.

In the case that no trajectory is found to fulfill the required conditions, another starting point is chosen at a distance in front or behind the vehicle, entering an iterative process to improve the success rate of the algorithm. Since, in a parking maneuver, a vehicle can move either forward or backward, each point in the trajectory has a direction value assigned, ψ=1 for the forward direction and ψ=−1 for the backward direction. Additionally, the trajectory is segmented considering the driving direction, so the transition is smoother.

Once a trajectory is chosen, it is necessary to represent it in the reference system of the vehicle and obtain equidistant points at distance $d_{s}=v_{d}\;Ts$, $v_{d}$ being the desired parking speed.

In the case of parallel parking, two tangent circular segments are defined to perform the parking maneuver, as depicted in Figure 9. The result is not restricted to a single trajectory, so the first feasible one is chosen as the reference. As in the previous case, constraints regarding steering and bounding boxes to avoid collisions are considered. Figure 10 illustrates an example regarding the aforementioned procedure.

#### 2.3.2. Parking/De-Parking Tracking Controller

An MPC-based controller was implemented to follow the trajectories defined for these maneuvers (Figure 11). Different from the platoon-following case, both velocity and steering are controlled by the same controller, which allows the parking vehicle to consider both variables in the maneuver.

To implement the MPC, a kinematic bicycle model was considered as defined previously. The MPC controls both the steering wheel δ and the longitudinal velocity reference $v_{r}\left(t\right)$, which are followed by a PID-based low-level controller (Figure 11).

Similar to the vehicle-following case, a quadratic cost function *J* is defined:(7)$$J={\hat{e_{p}}}^{T}Q\hat{e_{p}}+\Delta u^{+T}R\Delta u^{+}$$
where $\hat{e_{p}}={\left[\begin{array}{ccc} e_{p}\left(k+1\right) & \ldots & e_{p}\left(k+N+1\right) \end{array}\right]}^{T}$ is the distance between the vehicle and the reference position each time: $\sqrt{{\left(x_{k}-x_{refk}\right)}^{2}+{\left(y_{k}-y_{refk}\right)}^{2}}$; $\Delta u\left(t\right)={\left[\delta \left(t\right)v_{r}\left(t\right)\right]}^{T}$ comprises both steering and the longitudinal velocity reference.

The control problem takes the general form defined in Equation (Equation 4) in which the following input constraints are applied: *v* < 8.33 m/s, δ<0.7 rad, and δ>−0.7 rad.

### 2.4. Joining

Due to the high density in urban scenarios, it is possible that an external vehicle interrupts the formation of a platoon if the gap is too big. In order to reduce this effect, the follower should join the platoon as soon as possible once the de-parking has been executed. This is carried out by defining an additional state in the behavioral framework designed to quickly reduce the distance between a follower and the preceding vehicle by increasing its speed until a certain distance is matched. For this purpose, the CACC controller defined for the vehicle following state is used, in which more aggressive parameters are selected, while limiting the reference value to the speed limits in urban scenarios. Similarly, the lateral control is the same used in the aforementioned maneuver.

Note that this state will only be executed if the relative distance between the follower vehicle and the preceding one after de-parking exceeds a predefined value. If not, the state will directly change to vehicle following (Figure 2).

## 3. Simulation Framework

Control and behavioral planning algorithms are easily tested by coding them using an Integrated Developer Environment (IDE). However, there are limitations implied with this approach. Perception systems are key in automated vehicles since they are responsible for the positioning of the controlled vehicle and the object recognition. The information obtained by these algorithms, along with the messages received via communication are then used to make the safest decision that fulfills the current goal. Therefore, there is a need to build a virtual environment that can simulate all the platoon-related systems, so that it is possible to test and validate this technology safely and even provide an incremental approach of the design and test of the whole framework.

Thanks to the evolution of technology, 3D environment simulators such as rFpro [59], Congata [60], and Nvidias’ Drive Constellation [61], among others, have been proposed recently. Features such as dynamic lighting, weather change, or sensor simulations tied with state-of-the-art vehicle dynamic models and realistic assets have proven a great way of testing new autonomous-driving-related technology. Furthermore, the easy accessibility to game engines has made possible the release of some open-source projects, such as LGSVL [62] or the CARLA Simulator [63].

This work used the functionalities provided by the CARLA Simulator, in which a client–server configuration is used (Figure 12). The server handles the 3D urban open-world scenario defined in the simulation, along with all the actors present in it (objects such as traffic lights, roads, etc.), pedestrians, and vehicles. The behavioral planner framework and related functionalities are executed on a client, which interacts with the vehicles in the simulation. Each vehicle in the platoon was considered a single object, while had its control script, which is able to communicate with the rest of the vehicles by simulating the communication approach defined in the previous section. Additionally, the Global Navigation Satellite System (GNSS) virtual sensor provided by the Carla Simulator environment was used to obtain the positions of the vehicles.

Hence, the framework was designed then to manage a decentralized control system where each vehicle can make its own decisions based on the information received from both perception and communication. The ability to obtain kinematic information from the server makes it possible to bypass the perception algorithm for the sake of simplicity and allows defining a basis on which perception algorithms can be easily later integrated with the approach to test the overall approach.

## 4. Validation and Use Case

The relocation problem was simulated in the environment detailed in the previous section to test the validity of the proposed behavioral planner framework. The proposed use case has three actors take part in it: a leader vehicle and two follower vehicles.

The leader vehicle is supposed to be driven by a staff member of the car-sharing company. To emulate a human driver, the proposed MPC lateral and longitudinal controllers were used to follow a predefined path. The two follower vehicles are to be relocated. The first one is parked at point P1 and the second one at P2 (Figure 13). In addition, two other parking spots exist (P3 and P4). P1 and P3 are line parking spots, while P2 and P4 are battery parking spots. The global trajectory of the leader vehicle is depicted in blue and the starting position in green and was assumed calculated by a global planner, which is not the focus of this work.

For this simulation, the velocity of the leader vehicle was set to 30 km/h, which is known as the velocity limit in residential urban environments. The distance gap for the platooning was set to 7 m, considering this value represents the distance between the centers of the preceding and the following vehicles. A fixed step time of $T_{s}=0.05$ s was used in the simulations using the synchronization feature of Carla Simulator. The parameters of the MPC controllers detailed in Section 2 are gathered in Table 3.

The values of the PID controllers used for the CACC feature and the velocity following control are listed in Table 4.

The overall performance of the controllers can be seen in both Figure 14 and Figure 15. Figure 14a shows the velocity of both the leader vehicle and the two followers during the whole scenario, positive values meaning the vehicle is moving forward and negative values meaning the vehicle is moving backwards. Figure 14b, on the other hand, details the active states for each follower. Finally, Figure 15 illustrates the relative distance, measured point to point to the predecessor vehicle in the platoon, of the followers. In this case, a distance can only be measured when a predecessor is defined, either when a vehicle is in the joining or following states. Therefore, Figure 15 only shows values in the time lapses where the followers are in the platoon-following state.

As can be seen in Figure 14a and Figure 15, the errors of the first follower around steps 600 and 1050 were due to two sharp turns during the trajectory. This error will be higher as the curvature of the path increases. However, the controller used in this specific use case was able to overcome the error to follow once again the established velocity.

The picking-up process starts when the leader detects a vehicle is parked near its position at point P1 (Figure 16a). The message sent by the leader makes the follower vehicle change from a waiting state to a de-parking state (Figure 14 and Figure 16a). The resulting maneuver places the follower vehicle behind the leader one. However, as seen in Figure 15, the relative distance between both vehicles is higher than the defined threshold of 7 m; hence, a joining state is entered by the follower vehicle (Figure 14b), increasing its speed (Figure 14a) until the desired relative distance reference is achieved (Figure 15). This allows entering the following state, and the CACC longitudinal control is activated (Figure 16a). From this point on, the follower can maintain a relative distance from the leader. Slight deviations occur in steps 560 and 1020 due to the 90${}^{\circ }$ curves taken. Notice that, since the inter-vehicle distance is calculated point to point, the curves introduce a slight error, which can be neglected, as can be seen in the Figure 14.

The pick-up process of the second vehicle at point P2 is performed similarly to the first one, as depicted in Figure 14, Figure 15 and Figure 17. Note that the second follower takes the relative distance from its preceding vehicles, that is the first follower vehicle. In this case, as in the previous one, after de-parking, a joining state is required to place the follower at the desired distance in the platoon. This is depicted in step 104, where a velocity increase is required to catch up with the platoon after de-parking (Figure 14a).

Once both followers have been picked up, the platoon advances under the guidance of the leader vehicle. Note that, as depicted in Figure 15, the relative distances are properly followed by both followers.

The leaving process at P3 is performed similarly to the pick-up process. Note that, in the proposed approach, as all vehicles are similar, the last one in the platoon is supposed to disengage. In this case, the leader sends the message when it detects an empty parking spot near the area it is supposed to leave the relocated vehicles (Figure 18a). This way, Follower 2 modifies its state to the parking state, stopping the vehicle first and calculating the parking maneuver for line parking. Then, the parking trajectory control follows the defined maneuver, as depicted in Figure 19. Once the vehicle is properly stopped, it enters the waiting state; see Figure 14.

Finally, Figure 20 illustrates the same process for Follower 1, the last element in the platoon, in the case of a battery parking spot in P4. In this case, the trajectory calculated and executed by the follower in the parking state is depicted in Figure 21.

Based on the aforementioned simulations, it can be seen that the proposed framework has a proper performance, and hence, it constitutes an appropriate approach to further develop urban-based car-sharing applications.

## 5. Conclusions and Future Work

In this work, a behavioral planner framework was proposed for car-sharing applications in urban environments, which includes the definition of a finite state machine and a basic communication protocol. The proposed work aimed to provide insight into the development of new automated-vehicle-based solutions in urban environments for car-sharing applications.

In order to test the approach, a simulation framework based on the CARLA simulator was used to test a scenario in which a leader vehicle relocates two vehicles at a low speed, as if they were assets from a car-sharing business model. Car following, lane following, and parking maneuvers were validated for this purpose. In the simulation, the control framework performed correctly.

However, other information, such as traffic lights and vulnerable road users, could be considered. These can have a big impact on the state management of connected and automated vehicles and, thus, will be considered in further iterations of the proposed framework. Moreover, in this work, an optimal fleet relocation planner that defines the global path to be followed was assumed, which will be included in future work by the authors.

## Figures and Tables

**Figure 1 sensors-22-08640-f001:**
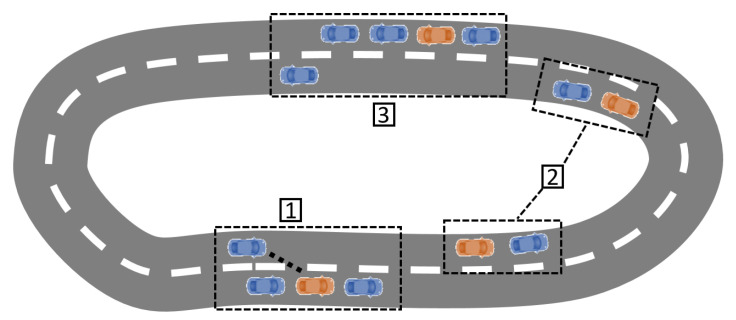
Relocation process. (1) The AV follower detects a leader. (2) A platoon is formed and drives together to the next parking spot. (3) The AV follower parks in the selected parking spot.

**Figure 2 sensors-22-08640-f002:**
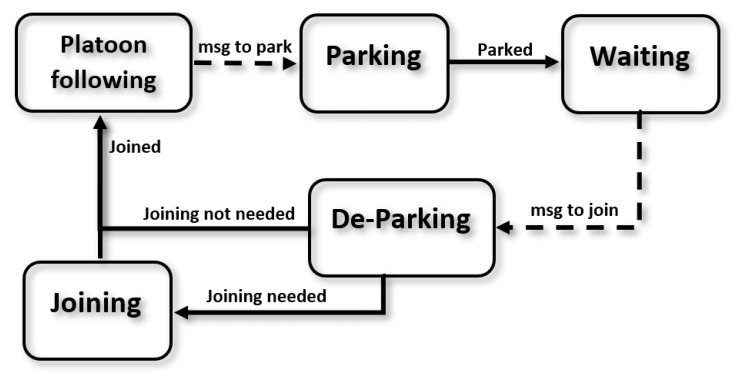
Finite state machine for the followers’ behavior. Communication-related jump conditions are represented with dashed arrows.

**Figure 3 sensors-22-08640-f003:**
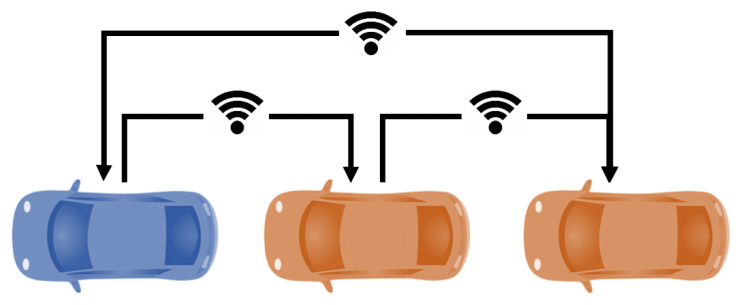
Platoon communication.

**Figure 4 sensors-22-08640-f004:**
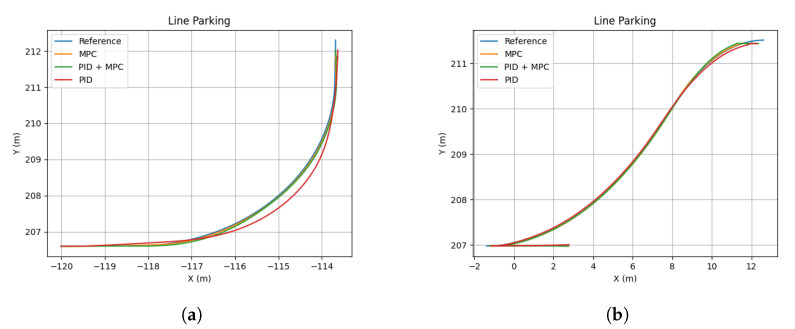
Parking track following controller comparison. (**a**) Battery parking comparison. (**b**) Parallel parking comparison.

**Figure 5 sensors-22-08640-f005:**
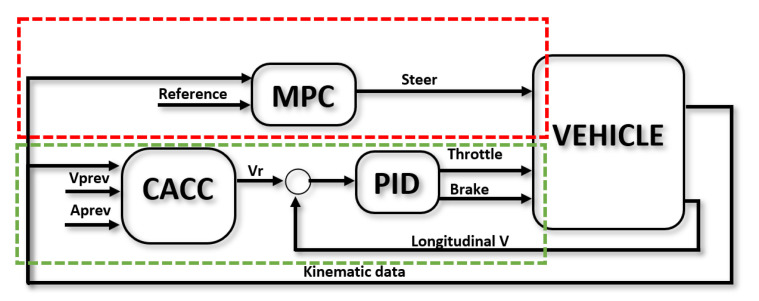
Control for platooning. Blocks inside the red area are related to the lateral controller, while the ones inside the green area are related to longitudinal control.

**Figure 6 sensors-22-08640-f006:**
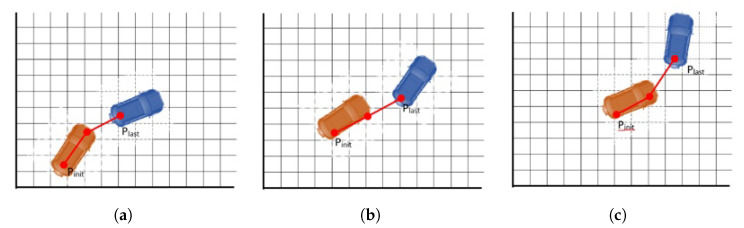
Path planning in followers (**a**–**c**).

**Figure 7 sensors-22-08640-f007:**
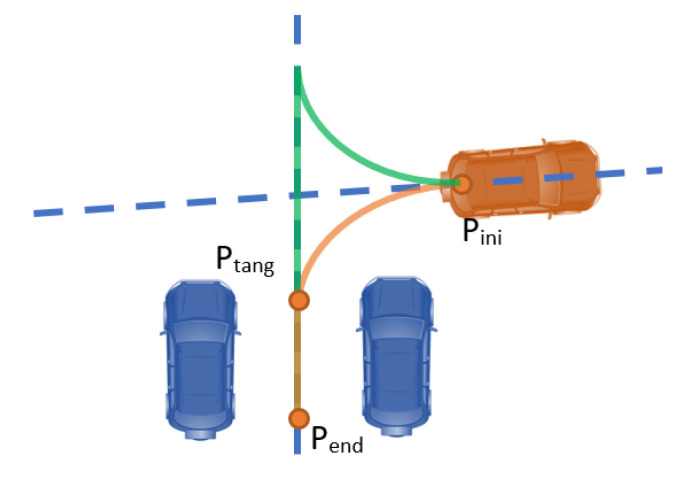
Battery parking trajectory generation example.

**Figure 8 sensors-22-08640-f008:**
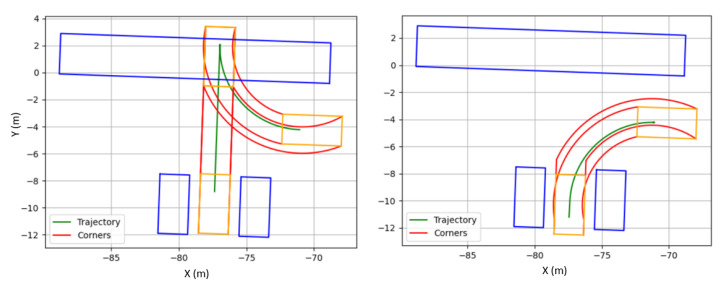
Battery parking collision check. **Right**: A valid trajectory. **Left**: a not valid trajectory. Blue boxes represent parked vehicles. Orange boxes represent the vehicle to be parked. Red lines represent the limits for the parking process, in order to detect collisions.

**Figure 9 sensors-22-08640-f009:**
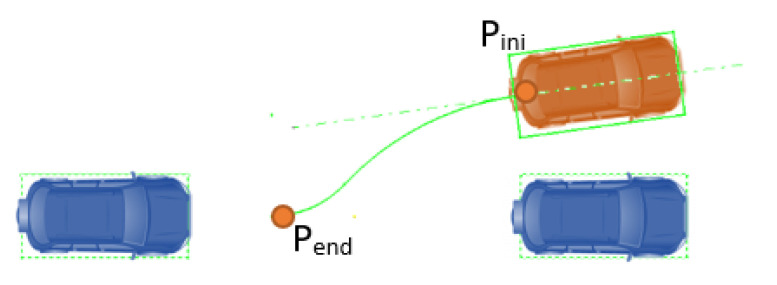
Parallel parking trajectory generation example.

**Figure 10 sensors-22-08640-f010:**
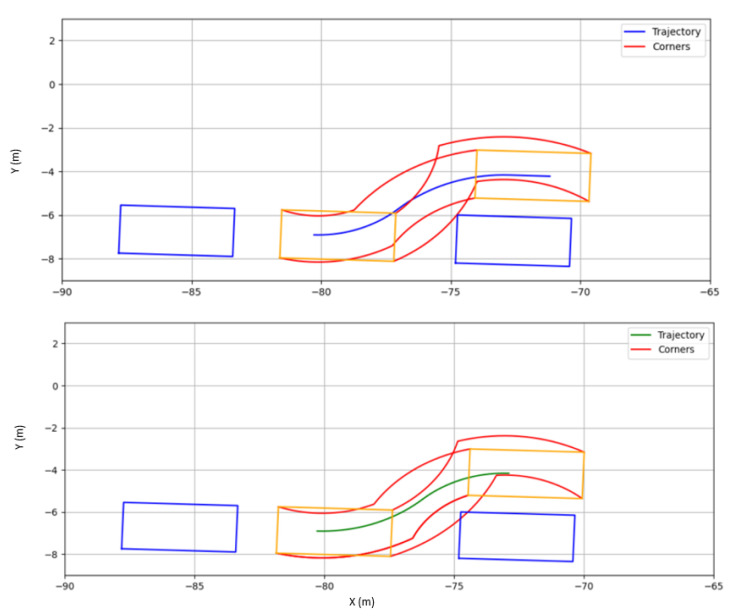
Parallel parking collision check. Up: A valid trajectory. Down: a not valid trajectory. Blue boxes represent parked vehicles. Orange boxes represent the vehicle to be parked. Red lines represent the limits for the parking process, in order to detect collisions.

**Figure 11 sensors-22-08640-f011:**
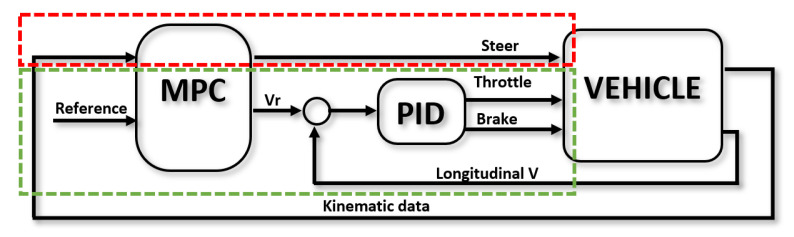
Control for parking. The MPC controller handles both lateral (red) and longitudinal control (green).

**Figure 12 sensors-22-08640-f012:**
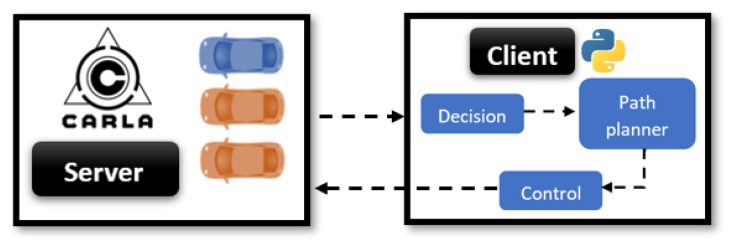
Carla Simulator client/server workflow.

**Figure 13 sensors-22-08640-f013:**
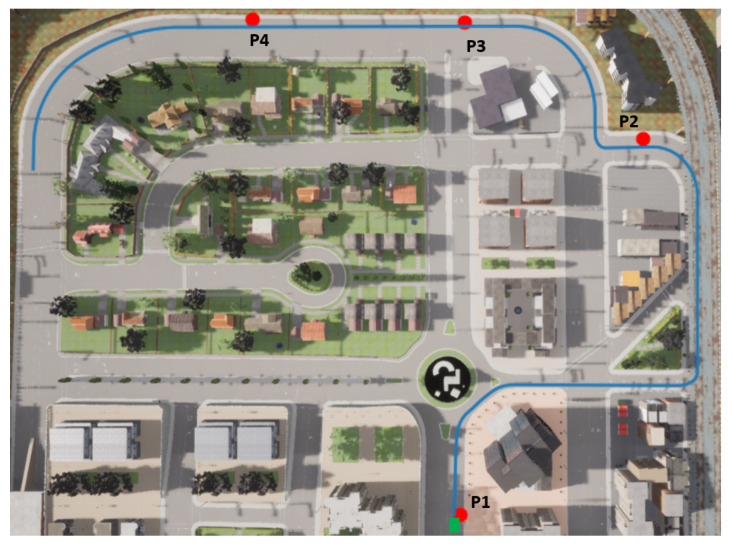
Use case scenario.

**Figure 14 sensors-22-08640-f014:**
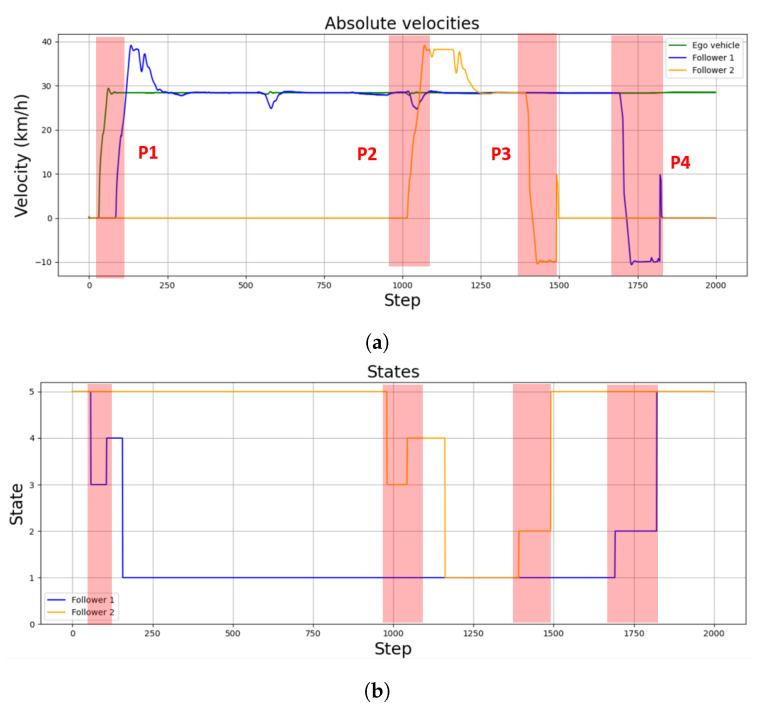
Velocities and states of the relocated vehicles during the use case. (**a**) represents the velocities of each vehicle. (**b**) represent the state of each vehicle: (1) Car following; (2) parking; (3) de-parking; (4) joining; (5) waiting. Parking and de-parking points are highlighted in red.

**Figure 15 sensors-22-08640-f015:**
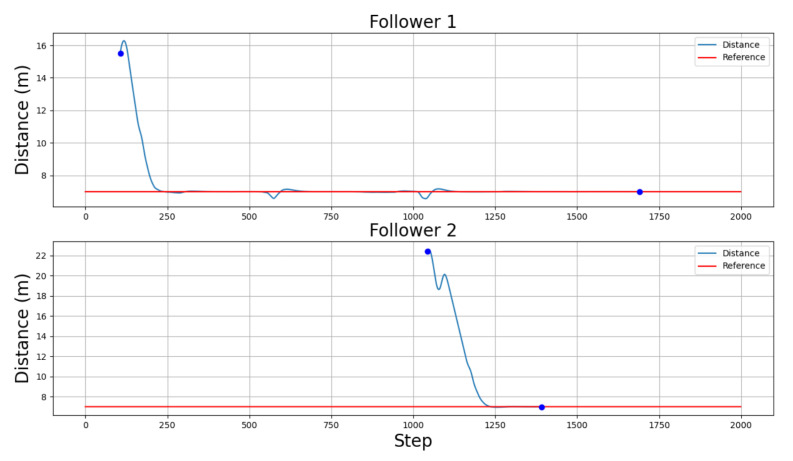
Followers’ distance to their predecessors along the simulation, but only represented during the platoon-following state time stamps.

**Figure 16 sensors-22-08640-f016:**
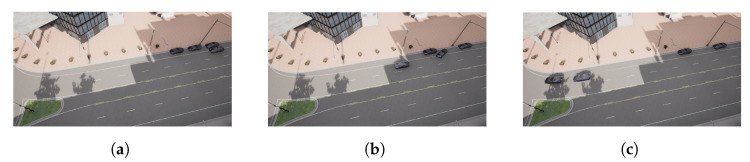
Follower 1 pick-up process. (**a**) Follower 1 waiting. (**b**) Follower 1 de-parking. (**c**) Follower 1 joining.

**Figure 17 sensors-22-08640-f017:**
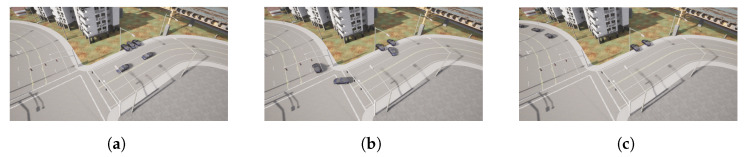
Follower 2 pick-up process. (**a**) Follower 2 waiting. (**b**) Follower 2 de-parking. (**c**) Follower 2 joining.

**Figure 18 sensors-22-08640-f018:**
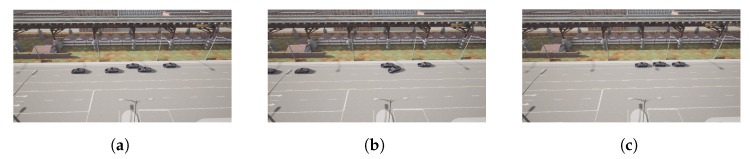
Follower 2 parking process. (**a**) Follower 2 in platoon. (**b**) Follower 2 parking. (**c**) Follower 2 parked.

**Figure 19 sensors-22-08640-f019:**
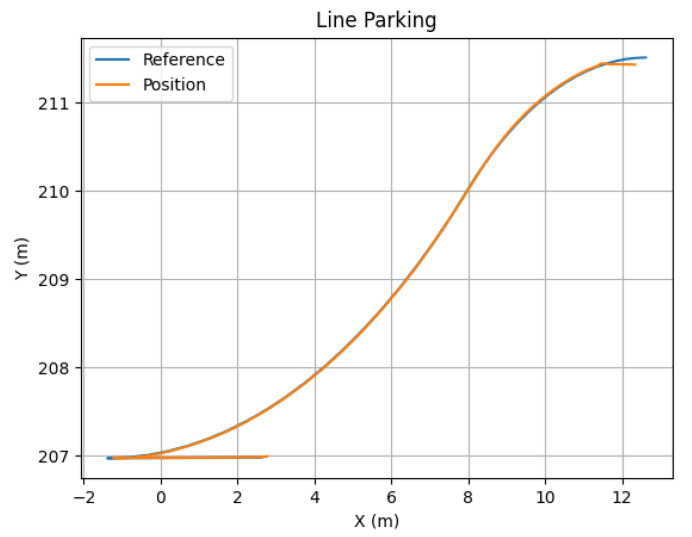
Position and reference in the line parking maneuver.

**Figure 20 sensors-22-08640-f020:**
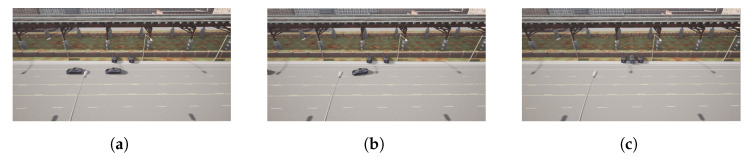
Follower 1 parking process. (**a**) Follower 1 in platoon. (**b**) Follower 1 parking. (**c**) Follower 1 parked.

**Figure 21 sensors-22-08640-f021:**
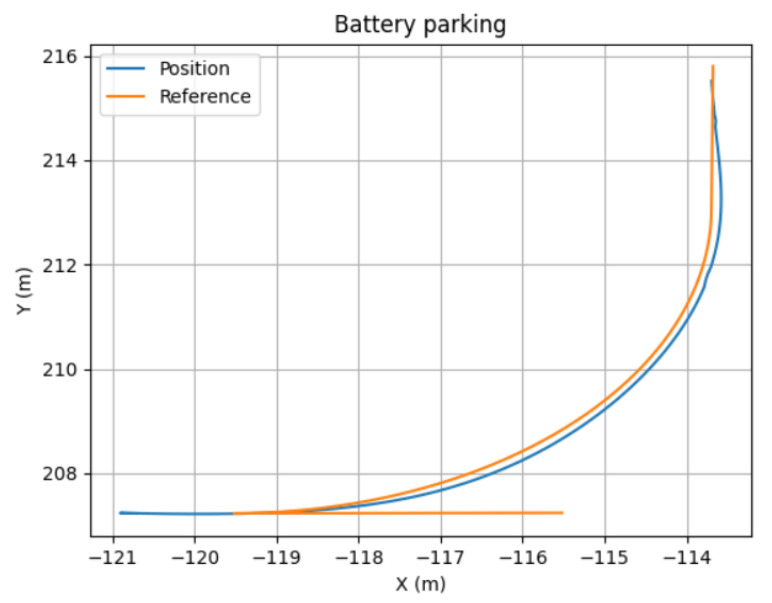
Position and reference in the battery parking maneuver.

**Table 1 sensors-22-08640-t001:** Controller comparison of RMS error for parking maneuver. PID-s (Kp = 10; Ki = 0.1; Kd = 0). Long. PID + Lat.MPC (Q = 100; R = 1.0). MPC (Q = 5; R = [0.001, 0.5]).

Controller	Battery Parking (m)	Parallel Parking (m)
Independent PID	0.072	0.024
Long. PID + Lat. MPC	0.064	0.018
MPC	0.023	0.012

**Table 2 sensors-22-08640-t002:** Control approach related to each maneuver (CACC stands for Cooperative–Adaptive Cruise Control).

State	Longitudinal	Lateral
Platooning/Joining	PID-based CACC	MPC (with inner PID velocity Loop)
Parking/de-parking	MPC (with inner PID velocity Loop)	

**Table 3 sensors-22-08640-t003:** MPC controllers’ parameters. st: steering weight, v: velocity weight (adimensional), h: prediction horizon (control steps).

	Following Lat	Parking Lat
**R**	st = 0.2, v = 2.0	st = 0.3, v = 2.0
**Q**	10	30
h	12	12

**Table 4 sensors-22-08640-t004:** PID controllers parameters. Units for CACC controller: $K_{p}$ (1/s), $K_{i}$ (1/s${}^{2}$) and $K_{d}$ (-). Units of Low level controller: $K_{p}$ (%s/m), $K_{i}$ (%/m) and $K_{d}$ (%s${}^{2}$/m).

	CACC	Low Level
** $K_{p}$ **	2	1
** $K_{i}$ **	0.5	0.2
** $K_{d}$ **	0	0

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
