# Peer review of "A Complete Framework for a Behavioral Planner with Automated Vehicles: A Car-Sharing Fleet Relocation Approach"

_sensors, 2022, doi:10.3390/s22228640_

Round 1

Reviewer 1 Report

This paper proposed a method for platooning and parking maneuvers for car-sharing application. This is based on a behavioral planner for car-sharing. The idea introduced by the authors is quite interesting. However, there are still some issues that need to be improved.

  1. In Line 102, it is expressed as “a novel behavioral planner ~”. I do not see in what sense there is novelty in the proposed method. If there is any novelty compared with other researcher’s results, this should be emphasized in the Introduction section.
  2. In Line 70, the definition of “CACC” is missing. This should be clarified before it is used.
  3. Regarding the parking/de-parking tracking controller, there are many autonomous parking study results. As it was referred in [14], can the automated parking and de-parking performance be compared for performance test? If not, the authors should compare some relevant and up-to-date references to prove the effectiveness.
  4. In Line 91, “according to that.” Here, “that” should be explained clearly.
  5. In Lines 128-129, To provide this functionality, a Finite State Machine (FSM) is defined in Figure 2. However, the “Finite” is not reflected directly or clearly in Figure 2
  6. PID control and MPC are adopted in this study. However, the stability proof of overall system was not given. Append all procedure of stability using mathematical equations.
  7. In line 199 and 200, the author notes that the error can be neglected in 'small step times combined with small velocities'. However, in Figure 13, the speed goes up to 40km/h. In general, this is shown at normal speed. Can you clarify this 'small velocity'?
  8. In line 175, the author referred two controllers, longitudinal and lateral control. However, in Figures 4 and 10, there is only longitudinal and no explanation on lateral control. This seems to need an explanation.
  9. In Figure 13, the position is not properly following the reference. Simulation needs to be improved.

Author Response

Dear Reviewer 1,

Thank you for the comments. We attach the responses in the pdf named "Rebutal Asier Arizala.pdf"

Kind regards

Reviewer 2 Report

Please refer to the following article:

Author Response

Dear Reviewer 2,

Thank you for the comments. We attach the responses in the pdf named "Rebutal Asier Arizala.pdf"

Kind regards

Reviewer 3 Report

Reviewer#1: In this work, the authors propose a new behavior planner of car-sharing applications for the relocation process of car-sharing business models. This method applies the vehicle platooning scheme to car-sharing relocation. While this approach sounds interesting, I still have several questions about this work as follows:

1) The abstract does not clearly indicate what the authors have achieved by mentioning the method.

2) In part 2, the authors mention that to implement the method proposed by the authors requires a global planner to define the optimal trajectory in advance, so how should it be defined?

3) It is suggested that the words depicting the 5 states of the vehicle in Figure 2 are consistent with the description in line 130 of the text.

4) The authors mention that there are two behavioral planner methods, so why did the authors use FSM but not NN?

5) What are the units of Figure 7, Table 1 and Table 2?

6) If two AVs are located very close to each other, how to solve the conflict between the two AVs during the merging process?

Author Response

Dear Reviewer 3,

Thank you for the comments. We attach the responses in the pdf named "Rebutal Asier Arizala.pdf"

Kind regards

Round 2

Reviewer 1 Report

The authors have revised the manuscript considering the suggested comments. Actually, some comments have been answered, but others are still not answered fully.  Still, the paper has some issues to be solved as  follows,

1. Although they proposed am method for platooning and parking maneuvers for car-sharing application, it is very hard if they used any kind of sensors. I don't think this paper does not match with the scope of this journal. On the contrary, it suits well with other journal such as 'Processes' or Applied sciences' on MDPI. Explain how this paper fits with the aims or research scope of this journal. And what kind of sensory or how this sensor is used or connected for realization.

2. In figures 13 and 14, it is still hard to read x and y label. Also legend. 

3. The step goes to 2000 in Figs. 13 and 14. What is the step size? To be able to implement, the authors should prove the time spent to realize the proposed algorithm.

4. In the previous comment # 3, it was pointed out that performance comparison should be done. Although the authors answered  by updating Section 2.1 and adding Table 1, it is not enough. Direct performance comparison in the simulation needs to be processed.

Author Response

Dear Reviewer,

We submit the answers in the attached pdf.

Thank you

Reviewer 3 Report

this paper can be accepted

Author Response

Dear Reviewer,

We submit the answers to the review in the attached pdf.

Thank you